# A Sociotechnical Approach to Analyze Pharmaceutical Policy and Services Management in Primary Health Care in a Brazilian Municipality

**DOI:** 10.3390/pharmacy9010039

**Published:** 2021-02-12

**Authors:** Noemia Liege Maria da Bernardo, Luciano Soares, Silvana Nair Leite

**Affiliations:** 1School of Health Sciences, Universidade Federal de Santa Catarina, PPGFAR Universidade do Vale do Itajaí, Itajaí 88300000, Brazil; liegebernardo@univali.br or; 2Department of Pharmaceutical Sciences, Universidade Federal de Santa Catarina, Florianópolis 88040000, Brazil; luciano.soares@ufsc.br

**Keywords:** pharmaceutical services, sociotechnical analyses, primary health care, pharmaceutical policy, pharmaceutical system

## Abstract

The decentralization of the Brazilian health system required that municipalities took responsibility for the local Pharmaceutical Policy and Services (PPS) system. This article presents and analyses an innovative experience of diagnosis of municipal PPS as a sociotechnical system. We adopted a multi-methods approach and various data sources. Sociotechnical theory was the framework of the methodology of evaluation and design of systems, analyzing the External System (health system, stakeholders, financing) and Internal System (goals, management, workforce, infrastructure, processes, technology and culture). The “objective” component of the PPS system was identified as the central element. The lack of a unified objective and of a central coordination and unmanaged pharmaceutical services prevented integrated internal planning and planning with other sectors. Stakeholders and documents referred only to technical elements of the system: Infrastructure, technical process, and technology. The social components of the workforce and culture were not mentioned. The organizational culture established was the culture of isolation: “Each one does his own”. The pharmacists working in the municipal health system did not know each other. There was no integration strategy between pharmacists and their work processes. Consequently, the municipal PPS had limited scope as a public policy. It had constrained the characteristics of PPS as a complex and open system. Understanding the municipal PPS as a sociotechnical system can push the development of a new level of policy and practice to ensure the population’s right to the access to and rational use of medicines.

## 1. Introduction

Access to medicines and its rational use are persistent global concerns. These issues have a major impact on the quality of the health system and, consequently, on the health outcomes [1]. The expansion of access to medicines was listed as one of the 10 biggest problems that demand attention from Work Health Organization [2]. A systematic review of rational use of medicines showed the inappropriate use of pharmaceutical products remains a public health problem [3]. It included 900 studies from 104 countries. Ensuring access to medicines and their proper use is a mission that requires coordinated action. Pharmaceutical policies cannot be based on either market decisions or collective common sense.

In Brazil, access is a mission defined as a public policy [4]. The National Pharmaceutical Policy defines services and responsibilities at all levels of governance of the Unified Health System (SUS) to guarantee access to medicines and pharmaceutical services. With the decentralization of the health system, municipalities took a “series of responsibilities that demand the mobilization of knowledge and technical, managerial, and political skills relative to pharmaceutical policy and services” [5]. These new responsibilities, especially from the beginning of the 2000s, demanded theoretical-methodological definitions and operational services in the health departments that were well described by Marin et al. (2003), and disseminated as the “pharmaceutical policy and services cycle”.

This “cycle” defines an interconnected set of processes, involving selection, programming, acquisition, storage, distribution, and use of medicines. That framework was crucial for establishing the importance and technical-scientific characteristics of a new field of public health in Brazil. In almost two decades, the role of pharmaceutical professionals and technical management in the access to and rationality in the use of medicines has been established [6]. The high level of access to medicines for chronic diseases in primary healthcare (PHC) achieved a prevalence of 94.3% in Brazil between 2013 and 2014 [7].

However, some important weaknesses can be identified in the prevailing theoretical framework used in recent studies on the organization of the pharmaceutical policy and services (PPS) system in PHC. Particularly, research reports highlight the lack of integration of policies and practices for the organization of pharmaceutical services with a comprehensive approach to the health-disease process [8,9,10,11].

Beyond the unequal access to medicines, gaps in technical issues for the institutionalization of PPS in PHC were also uncovered. They included low availability of PPS manager jobs; few pharmacy and therapeutics committees and few lists of essential medicines; poor structuring of pharmaceutical services; low availability of computerized systems; limited logistics, management, and pharmaceutical care management in health facilities; limited integration between pharmacists and the health team; limited intersectoral governance; low participation of PPS managers in social control councils [10,12,13,14]. All this weaknesses described above were identified as technical problems, in a perspective of the pharmaceutical policy and practice as an independent part of the health system, and a techno-focused field. The personal and social factors influencing the field are currently underestimated in the studies.

This scenario outlines the premise that the complexity of the PPS requires its understanding as a complex system. Its functioning is best explainable from an interconnected multidimensional perspective. With this in mind, this article analyses an in-service experience of a municipal PPS diagnosis from the perspective of sociotechnical system. The theory of sociotechnical systems is a contemporary way to provide support for various aspects of technical and human activities in organizational development. It consists of interdependent subsystems, organized as an open system that interacts with the external environment and with its own internal environment [15]. It first characterizes organizations as sociotechnical systems made up of two subsystems, the technical and the social, which work together to accomplish the organization’s task or mission [16].

In this study, sociotechnical theory was the method framework for the evaluation and design of systems, from the classic studies by Trist (1981), developed by several authors who applied the theory in different sectors. In health systems, this theory has been used to evaluate strategies for incorporating technological innovations [17,18,19,20], and in pharmaceutical services [20,21]. However, this is the first approach of the sociotechnical systems theory to characterize and analyze the PPS of a municipality in Brazil, aiming to prepare the interventions plan.

## 2. Materials and Methods

We conducted the study during the situational diagnosis of PPS of a municipality. The diagnosis was a step in a project to include the pharmaceutical field in a PHC inter-professional collaboration residency course. It reports an in-service experience to develop a sociotechnical assessment of PPS in PHC. The diagnosis design aimed to promote the reorientation of the system. We reported the diagnosis results from one of the four municipalities participating in the project. The study had the partnership of the Municipal Health Secretariat (MHS).

### 2.1. The Place of Study

The municipality is located in the European Valley, in the state of Santa Catarina/Brazil, and has an estimated population of 135,000 inhabitants. The Human Development Index (HDI) is approximately 0.8, and the main economic activities are industry, agriculture, and services [22].

### 2.2. Data Collection

The search in different data sources helped to reach the scope of the interpretation in a sociotechnical analysis. The collection covered data until June 2017.

#### 2.2.1. Sources, Search Strategies, and Data Characterization

The extraction of sociodemographic and financial data took place in public databases available on the internet. The variables collected from each source are in Table 1.

**Table 1 pharmacy-09-00039-t001:** Data extracted from consulted databases.

Database	Link	Description of the Data
IBGE ^1^	https://cidades.ibge.gov.br/	Year of Installation, territorial area, distance from the capital, estimated population, human development index, predominant economic activity, GDP per capita, percentage of urban households [22].
SIDEMS ^2^	https://indicadores.fecam.org.br/index/index/ano/2020	Municipal sustainable development index (IDMS), health situation analysis [23].
Municipal Transparency Portal	https://brusque.atende.net/?pg=transparencia#!/	Expenses with pharmaceutical policy and services [24].

^1^ IBGE (Brazilian Institute of Geography and Statistics); ^2^ SIDEMS (System of Sustainable Municipal Development Indicators).

Other sources included the chapters “Structure of the health care network” and “Pharmaceutical Policy and Services” of the Municipal Health Plans of 2010 and 2014, municipal guidelines for PPS, the municipal controllership evaluation report, Government Plan of the mayor elected in 2016, and the municipal PPS regulations until 2017. Municipal Health Plans are instruments that define the municipality’s commitments to Unified Health System (SUS) management and are required for the municipality to receive funds from the federal system financing responsibility. Strategy involved exhaustive systematic search and document analysis. The collected data included:
Municipal health system: Infrastructure, organizational structure, Family Health Strategy coverage, workforce description of the Family Health Support Center, health funding, and financial circumstances;PPS: Goals, organizational structure, available workforce and infrastructure, processes, and organizational culture.

An institutional documentary published on the proposal for reorienting PPS in the municipality provided testimonies from managers, pharmacists, coordinators of health units, doctors, and patients of municipal health services. The testimonies were collected in 2018, after the projects were implemented. So, they inform about the viewpoint of the participants about the PPS situation until 2017 [25]. Other sources of data were newspaper articles on issues related to the PPS published in the city between January 2016 and February 2017, radio stations news, and internet blogs’ information about the municipal PPS. The search strategy combined the terms “medicine” and “pharmaceutical policy and services”.

#### 2.2.2. Assessment Matrix of the Management Capacity of Municipal PPS

We used the Assessment Matrix of the Management Capacity (AMMC) instruments to collect the municipal PPS data. Manzini and Mendes [9] developed and validated AMMC in PPS of Santa Catarina municipalities. The matrix framework uses Carlos Matus’ management assumptions, first adopted by Barreto and Guimarães [8].

The collection was performed at the pharmacies from the municipal health department. Among the collection sites, one isolated pharmacy dispensed PHC medicines and another pharmacy provided specialized medicines and some medicines dispensed under judicial process. Other sites included 23 pharmacies located into PHC centers, a pharmaceutical center for distribution and supply, and the pharmacy located in the Health Municipal Secretariat. Participants of this phase were: One health secretary, one PHC Coordinator, one pharmacist, 17 nurses/coordinators of the PHC Centers, 18 doctors, 18 pharmacy assistants, and 86 patients.

### 2.3. Data Analysis

The documents evaluation was performed by content analysis. The social and technical components of the municipal PPS system were typified according to the sociotechnical framework described in Figure 1.

The model components were chosen after researching the literature on the analysis of sociotechnical systems in the health field [17,19,20,21,26,27,28] and in other sectors [16,29,30,31,32,33,34], and about PPS organization in Brazil [9,10,26,35,36]. PPS is represented in its external environment composed by the management of the municipality and the health system, and characterized by components of its internal system that modulate the process from the input (Prescription of medicines) to the outcomes (Access to and rational use of medicines).

The results were stratified in the socio-technical dimensions and typified based on the characteristics of the components described in Table 2. The meanings provided by the components allowed the different aspects of the PPS reality to be reframed, considering the complexity inherent in the sociotechnical nature of this system.

### 2.4. Ethical Aspects

This study is part of a project called “Application of Sociotechnical Theory in the Reorientation of Pharmaceutical Care in Primary Health Care”. The Univali Research Ethics Committee approved it (CAAE: 28471320.2.0000.0120). All participants signed a written form of consent after having received information about the study. We do not identify the municipality by name to prevent identifying individual participants. Individual participants were indicated by professional title and job position.

## 3. Results

The results of the socio-technical evaluation of PPS in PHC will be presented below in two sections: 1—organization of the environment in terms of stakeholders, objective; financial circumstances and health funding, the general regulations and those related to PPS system performance; and 2—description of the of socio-technical elements of PPS in PHC.

### 3.1. Organization of Environment

In 2016, the sustainable municipal development index (IDMS) was 0.726 (scale from 0.00 to 1.00), a medium high level. In the economic and social participation dimensions, the levels were low (0.630) and medium low (0.524), respectively. The analysis of the health situation falls within a medium high level, with an index of 0.798. In 2017, the municipality experienced weaknesses in the economic and political contexts, attributed to the impeachment of the mayor and the vice-mayor in 2015. This scenario was accompanied by a strict control process, especially in the acquisition of products, and in the hiring of services and staffing by public management, with a financial surplus at the end of 2016 [24].

The main stakeholders identified in the municipality’s PPS were the social control council, the municipal health department, and the press. The Municipal Health Plan (2014–2017) embraces the guarantee of the dispensation of medicines from the Municipal List of Medicines (called REMUME) and, secondarily, building a laboratory to produce herbal medicines. Table 3 illustrates expectations and demands about PPS based on the summary of reports published in local media outlets. The demands regarding the results to be produced by PPS focus on the availability of medicines, given the understanding that the system failed to meet this expectation.

**Table 3 pharmacy-09-00039-t003:** Narrative from articles published in the municipal media about access to medicines in the Municipal Health Network.

Date	Headline	Narrative	Source
1/26/16	Lack of medication	“The article on the cover of the newspaper Município Dia a Dia, last Thursday (21), speaks of the lack of medicines inpublic pharmacies. According to the report, there are 53 drugs missing from the downtown basic pharmacy, and the list contains drugs prescribed very frequently, as is the case of the omeprazol, amoxicillin and buscopan”.	[37]
5/27/16	City Hall clarifies about lack of medicines in the Municipal Health Network	“The Health Secretariat informs that some medications are missing in the Primary Health Centers and in the Basic Pharmacy, located in the Center. The items have already been requested and the situation can be regularized at any time. According to the folder, the situation occurs due to several situations, such as lack of raw material for production, discontinuation of imports and lack of supplier. However, some of the remedies await only the supply of the bidding company”.	[38]
9/19/17	Patients suffer from lack of medication to relieve back pain	“Patients who have a back problem, suffer from severe pain and need Tramadol to relieve symptoms since the last week, face the shortage of medication.	[39]

Regarding the financing of PPS in 2017, the budget was R$5,245,270.18. Data from the municipality’s transparency portal show that between 2014 and 2017, actual expenses with PPS represented, respectively, 4.5%, 3.8%, 4.3%, and 4.7% of the expenses incurred by the municipal health system, which grew at a constant rate in the period. PPS payments increased from R$ 3,110,001.97 in 2014 to R$ 4,040,022.63 in 2017, varying from R$25.98 to R$ 31.36 per capita. The 2016 management report points out that “considering the high consumption of medicines by patients, this policy [PPS] was insufficient to reach its totality in the face of a lack of resources”. The 2017 report described a balance of R$ 366,312.02 not spent on PPS. The program and action expenses section of the municipality’s transparency portal shows that in the action “Family Health Strategy and Pharmaceutical Assistance”, the percentage of the executed budget was 99.68% in 2014, 96.70% in 2015, 93.88% in 2016, and 84.16% in 2017.

The municipality’s 2015 Annual Health Program (AHP) defined the “Guarantee of pharmaceutical assistance within the scope of SUS” in its Guideline H translated into objective H1 of “Ensuring the availability of medicines to the population” with the following planning, described in Table 4:

**Table 4 pharmacy-09-00039-t004:** PPS goals, indicators, actions, and budget in the 2015 AHP in the municipality.

Goal	Indicator	Action	Budget (R$)	Budget Origin
Implement actions to dispense medication and inventory maintenance regularly	Number of actions implemented	Take actions to ensure adequate dispensing of medication; Promote regular stock maintenance; Ensure adequate distribution in the municipal health network.	5000.00	Municipal resource.
Maintain the supply of medicines regularly.	Number of pharmacies in operation	Ensure the distribution of selected drugs on a regular basis; Regular stock maintenance; Adequate distribution in the municipal health network	2,034,534.40	Federal resource. Municipal resource.

Source: 2015 Annual Health Program in the municipality, p. 29 [40].

Municipal PPS regulations, published until 2017, were related to:(a)Complementary Law No. 224/2014: Institutes the Municipal Health Code, which defines pharmacy as a health service and medicines as products subject to sanitary control [41].(b)Normative No. 005/2015: It defined the organization and assignments of the tasks, the prescription, dispensing and supply of medicines, and the parameters for the functioning and structure of the services [42].(c)Decree No. 7826, 8 July 2016: Disciplines procedures to be adopted by doctors and dentists, municipal civil servants, and service providers for the Unified Health System at the municipal level in the prescription of medications [43].

### 3.2. Socio-Technical Characterization of the Municipal PPS

#### 3.2.1. Goal of the Sociotechnical System of PPS in PHC

The data collected were described and referred to by several stakeholders that affected the social and technical factors of PPS, as shown by the evidence below.

One promise of the mayor-elect campaign (2017–2020) was to increase the accessibility of medicines in health facilities. The government plan carried the slogan “Medicines basic and of chronic use to the entire needy community”. In the municipal health plans (MHP) for the periods 2010–2013 and 2014–2017, the objectives focused on the accessibility of medicines, as described in Table 5.

#### 3.2.2. PPS Management in PHC

In 2017, the municipal PPS did not have a formalized organizational structure, there was no coordination of the system, nor the formation of a PPS team. Six pharmacists work in services and with tasks considered independent. A representation of this organizational structure is shown in Figure 2 to illustrate the nature of administrative relationships.

Table 6 describes managerial and sociotechnical dimensions of the Management Capacity Assessment of municipal PPS in 2017.

#### 3.2.3. Workforce of Municipal PPS in PHC

Ten indicators of Assessment Matrix of the Management Capacity of municipal PPS correspond to sociotechnical dimension “workforce”, according to Table 6. Coordination was the responsibility of two nurses and an administrative assistant. Six pharmacists were active staff of the PPS, two with a fixed-term employment contract and four civil servants. The experience of pharmacists in the municipal health service ranged from to five years.

Nursing technicians and assistants were the major workforce to dispense medicines in health facilities. There was no predefined scale to work and the premise was “whoever is out of activity stays at the pharmacy”. The health services coordination (usually a nurse) asked for medicines from the warehouse. In the Central Pharmacy Dispensing Unit (UD1), there were two pharmacists and six assistants (a nurse and five administrative staff). These pharmacists planned PPS services in an independent way.

We found a pharmacist in the dispensing services in specialized care service center (UDI2). Two pharmacists and three assistants in specialized and judicial dispensing centers (UD2). One pharmacist worked at the Pharmaceutical Distribution and Supply Center. In all these services, pharmacists planned PPS activities themselves. None of the pharmacists had previous training to work in the public service or in PPS, nor have they received training in the area in the past two years. The quotes below show PHC professionals opinions about the pharmacists qualifying the healthcare processes:

PHC manager: “When I was at the PHC, we didn’t have this process in place yet, I can’t say it was a mess because it was part of the process, but we have two or three technicians in each health center and each one who had a little time at the pharmacy and dispensing medication, with that we had a lot of puncture in the stock, there were several complications”.

Family Health Strategy Doctor: “I believe that a pharmacist fixed in the PHC center would help us a lot, due to the knowledge, you know, for being a qualified professional for that. Although we have a technician in a finished shift, but we do not have the technical knowledge of that job, which has to be performed”.

#### 3.2.4. PPS Infrastructure and Processes in PHC

The infrastructure component has three indicators described in Table 6. We analyzed 22 UDI in PHC, UD1 (PHC medicines), and the Pharmaceutical Distribution and Supply Center. All facilities had internal and external areas in good conditions and hygiene. The structure offered no risk to patients and employees, meeting the recommended standards. We found computers and internet access in all units. Fifteen percent of facilities had insufficient number of computers. These computers had no technical maintenance for two years. An information system manages the stock of medicines in PHC. A third party company provides the system.

The processes component include 23 indicators. It was detailed in Table 6. In 2017, PPS services performed 20,406 attendance, as shown in Table 7.

In 2017, PPS services included scheduling, ordering purchases, storing and dispensing medicines. The procurement department of the city hall held bids to buy medicines with no pharmacist participation. We observed no services related to: (1) Discard health care waste (there is no Health Services Waste Management Plan); (2) review patients’ pharmacotherapy or perform pharmaco-therapeutic follow-up; (3) provide technical support for the health team or health education for patients and the community. There were no standard technical criteria for programming, purchasing, distributing, and dispensing drugs. Only in Central Pharmacy Dispensing Unit (UD1) did we find defined routines and procedures. There were no procedures related to pharmaceutical care and support from pharmacists for healthcare teams.

Pharmacists inferred the lack of standardized technical parameters can compromise availability of medicines. The availability of drugs was only 60%, and there was a large amount of drugs out of date. Several actors point to other deficits in PPS services. They attribute to this: Problems in the work process, waste of products, lack of access, and treatments with inferior quality. The quotes below illustrate the statements.

Secretary of Health: “... the current management found primary healthcare with ‘unstructured teams by the lack of professionals’, lack of medicines—60% of the items listed in the Municipal List of Medicines, 500 kilos of expired medicines and an empty warehouse”.

Pharmacists from Pharmaceutical Distribution and Supply Center:

“When I started at the warehouse, our purchases were based on the transfers that we made to the healthcare centers and then these values, these amounts that we had, they were not very reliable, because they were not based on the real demand that we had in the units at that time”.

“I worked in the health store, in the medication sector and in the period 2014–2016, we did not have stock control, we did not have the management of stock control in health units, nor which distributions in the stockroom. The distribution was made to health units once a month and that was the supply that the units have”.

Family Health Strategy doctor: “We had a very serious problem, in terms of user access, the medications in the unit, a difficulty due to the lack of professionals to take care of the medication release and this influenced the entire work process of other professionals”.

PHC center Coordinator: “So three years ago when I started in the municipality, at the health unit, PPS was very deficient in many ways, in the sense of stock control, waste, user guidance, professional guidance, that we didn’t have a lot of accessibility to some information and guidance regarding the delivery of the medication and PPS”.

PHC Manager: 

“When I was at the healthcare center, we didn’t have this process in place yet, I can’t say it was a mess because it was part of the process, but we have two or three technicians in each healthcare center and each one who had a little time went into the pharmacy and dispensed medication. With that, we had a lot of puncture in the stock, and there were several complications like that. Even for the nurse, it was difficult to charge someone who was a continuous process and walked smoothly”.

#### 3.2.5. Technology in PHC

The technology component consists of five indicators, detailed in Table 6. It includes drug treatment and the instruments for its access and monitoring. The municipality regulates the prescription and dispensing of medicines. It defines facilities to provide pharmaceutical services, as indicated in Figure 2. The municipality’s PHC Medicines List had the National Essential Medicines List as reference. In 2017, 202 drugs were available. About 16% (33) did not appear in the 2017 edition of the National Essential Medicines List. Some of these items had no prescription for over a year.

There was a municipal program for access to medicines, which were not selected by PPS. Patients should submit a medical request based on scientific evidence to have access. This program allowed access to 148 selected medicines. Most of them were outside the National List of Essential Medicines. However, such products had therapeutic equivalents on the municipality’s Medicines List. The program cost was R$540.00 a year. The municipality’s attorney analyzed the medicines lawsuits, without the participation of the pharmacists or the Health Department support. We observed some constraints on the instruments used in the municipality’s PPS:(a)Municipality’s Medicines List: Edition not reviewed for five years, not approved by the Municipal Health Council, and not published in an official bulletin. There were no defined procedures for disclosing the list.(b)Computerized PPS management system: Outsourced service, restricted to inventory management in PHC.(c)Electronic medical record: Pharmacists had access to patient data in all PPS facilities.(d)Standard operating procedures: Outdated and used only by Central Pharmacy staff.

#### 3.2.6. Organizational Culture in PHC

The organizational culture has four indicators (Table 6). The pharmacists working in the municipal health system did not know each other. There was no integration strategy between pharmacists and their work processes. The organizational culture has four indicators. Some pharmacists working in the municipal health system did not know each other. There was no integration strategy between pharmacists and their work processes. Some evidences of these we can found in pharmacists citations.

Pharmacist of Central Pharmacy: “When I started working here …, there was no PPS system, we worked each pharmacist in his workplace. We didn’t have any contact between us, some of us didn’t even know each other and that made the job very bad and very unrelated”.

Pharmacist of the Specialized Care Dispensing Center said “Specialized service has always been a very isolated sector of PHC”.

PHC workers expressed dissatisfaction with the lack of support for PPS actions. The nursing team managed the stock beside delivering medicines to patients. This caused overload for them. The following quotes illustrate this point.

Pharmacist in Specialized and Judicial Dispensing Center: “Until 2017, the municipality did not have a policy for PPS, we had pharmacists working one in each pharmacy, in the PHC pharmacies, in the out, in the specialized pharmacy, each one worked individually, not as a team or a group”.

Secretary of Health in 2017 said:

“When I arrived and took over the portfolio of the health department, PPS was quite confused, there was no line or better a municipal policy for PPS, and this is what we need today,. We are not only ensuring insuring costs or expenses, but working with the public money in a responsible way. So, we had a very messy house, a lot of expired medicines and put in stock, something that today we prioritize the right purchase, good purchase, right and something that each health center worked on dispensing medication in the way that best suits them”.

## 4. Discussion

The results of this real life project revealed the complexity involved in a PPS system. A Municipal PPS is characterized as an open system and interrelates with the municipal management and the municipal health systems as described in Figure 1. Within the sociotechnical approach, these larger systems represent the environment of the municipal PPS.

Despite the political weaknesses, the studied municipality showed good economic development. Its health indicators have a high average level. Social participation was the only indicator that was below average [23]. We can better understand some aspects of the sociotechnical nature of PPS in this context in which the stakeholders (health professionals, managers, and patients) were not pleased with the results of the municipal PPS.

The aim of the organizational structure was identified as the central element of the model. The other system components interrelate with its scope. For municipal managers, health professionals, and patients, the lack of medicines was the main frustration. The central aim of the Municipal Health Plan and PPS was the same. So, the aim of the PPS was “to have medicine available from the Municipal List of Medicines in the dispensing centers”. This aim exposes a conceptual reduction of the aim of one PPS sector in a public health system. A broader perspective comes from WHO: Availability is a dimension of access to medicines. So are affordability, acceptability, and rational use of medicines [44,45]. In Brazil, access is the purpose of PPS and involves its many dimensions [4]. It is more complex than the mere availability of medicines.

Nonetheless, medicines availability is a key part of the PPS architecture. This dimension can suffer many constraints. We need supply-side strategies to ensure medicine availability and help expanding access to medicines in health care systems [46]. However, when a PPS system chooses medicine availability as a central goal, its commitment is narrowed down to delivering medicines [6,8].

We identified weaknesses in the PPS management component. Lack of central coordination, pharmaceutical services with more than a manager, and a fragmented organization prevented integrated planning with other sectors (Figure 2). At the health secretariat, there were no human resources available to plan the actions that involved the services. PPS issues were discussed and decisions were made with no integration of the directorates (PHC, specialized services, or warehouse). In 76% of Brazilian municipalities, PPS coordination is nominated in the health department organization chart [10].

PPS has a management logic based on an input distribution model. The environment refers only to technical elements of the system: Infrastructure, technical process, and technology. The social components of workforce and culture are not mentioned in the Municipal Health Plan. However, the organizations’ reality is based on social components. People are essential sociotechnical elements, as well as the components affecting them [47]. The architecture of municipal PPS should not ignore general and organizational culture.

We did not find working groups organized in the municipal PPS. The working group is considered the elementary construction of the sociotechnical system. Self-regulation, semi-autonomy, and specific functions, but with interrelated tasks, are working group features. What connects the groups is a common goal. This increases the capacity to meet the demands of the internal and external environments [31].

The organizational culture established was the culture of isolation: “Each one does their own”. People working in the health system did not know each other in a municipality system with only 130,000 inhabitants. There were no shared symbols or collective rituals. The ideology was based on the care logic of contribution. The national pharmaceutical policy advocates PPS with a comprehensive approach to the health-disease process [8,9,10,11].

The distribution and dispensing centers’ infrastructure is the indicator that showed the best result. The result is better than those observed in other municipalities in the state [48]. However, the infrastructure does not seem to be enough to provide articulated pharmaceutical services. This component is connected to the PPS aim, restricted to distributing medicines.

The processes of the municipal PPS emphasized medicine supply. We highlight the regulation of technical processes for making medicines available. It involved prescription, acquisition, inventory control, and dispensing. Even with technical components in focus, pharmacists did not take part in planning or executing scheduling and purchasing services. In most municipalities in the state, this process involves the pharmacist. This reality affects availability, reasonableness of costs, and sustainability of the access to medicines. In view of the volume of financial resources allocated, this aspect is fundamental [48]. Processes that need intersectionality and participation, such as a Pharmacy and Therapeutics Committee and participatory planning, have not been carried out either.

The PPS technology component at PHC did not meet the needs of prescribers or patients. The problems involved both the diversity of medication available and their recurrent scarcity. Here the key factors are: The formal constitution of the Pharmacy and Therapeutics Committee; and the consumption of 10% of the budget, but the “non-selected medicines’ program”. In both situation, there were no defined access criteria. These factors are like those found in the studies by Hoepfner and Gerlack [10,48].

We can understand the weaknesses of the municipal PPS by looking at some results obtained by the system. The resource applied for the PPS (R$38.90/inhabitant) is higher than the national or state average (R$20.00) [49]. Even with a high cost, the observed effects show professionals’ and users’ dissatisfaction, as well as a waste of medicines.

What works in one configuration may not work elsewhere. We must adapt improvements to the local context and check. The sociotechnical systems approach is capable of answering to local specificities. It also helps to produce incremental results and contributes to building flexible organizational structures [17,31]. Understanding the municipal PPS as a sociotechnical system contributed to developing an intervention project. For Carayon et al. (2011), such understanding has the ability to transfer and put in place new knowledge and methods to influence the entire health system [47].

The results presented here suggest that the municipal PPS is a complex system, like the one Appelbaum found [30]. In a sociotechnical intervention, there are many strengths. They should be used within a strategic plan for organizational development instead of as an isolated approach. Policymakers and managers should target their support to systemic solutions, rather than contributing to proliferate fragmented efforts.

Some limitations of this study need to be highlighted. One difficulty is collecting data from different and poorly systematized sources. The lack of good electronic systems and transparency of public data weaken the ability to study this field. The scarcity of references to sociotechnical systems applied to PPS required the adaptation of study instruments, such as the management capacity assessment protocol, besides using references from administration and sociology fields.

## 5. Conclusions

The study revealed that the municipal PPS implemented had limited scope as a public policy. It had constrained the characteristics of a complex and open system. Stakeholders understood PPS as a set of technical procedures, without planning or integration. Its absence in the health secretariat organization’s chart symbolizes the poor understanding of the system in its policy dimension.

The municipal PPS was a fragmented set of operations and produced unsatisfactory results. The aim of the PPS was focused on the availability of medicines only; this subverts the logic of a system whose purpose is to provide access to medicines and its rational use. PPS was reduced to the availability of the input, with a low capacity to promote advances in health care. Even if pharmacists had technical-scientific training for developing clinical pharmaceutical services and go further, the situation found in the municipality did not offer the minimum conditions for doing so.

PPS has had a great development in all levels of government in Brazil in the last twenty years. A turning point strategy to understand municipal PPS as a sociotechnical system can push the development of a new level of policy and practice to ensure the population the right to the access to and rational use of medicines. It will allow advancing its contribution to the health care progress.

## Figures and Tables

**Figure 1 pharmacy-09-00039-f001:**
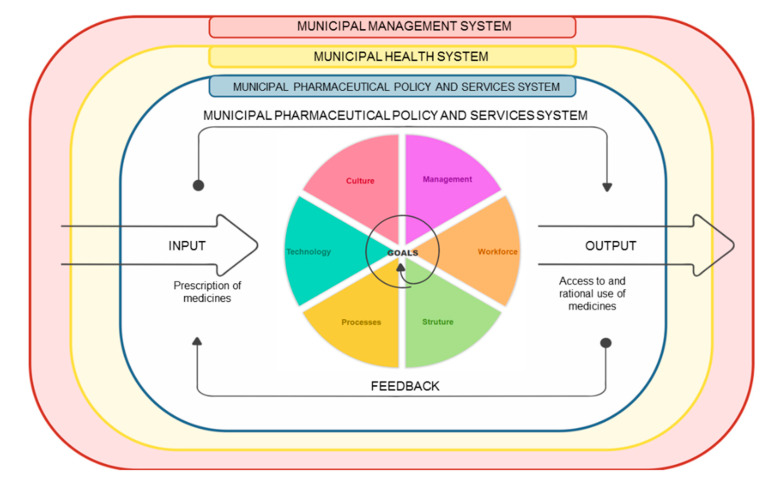
The social and technical components of the municipal Pharmaceutical Policy and Services (PPS) system. Source: The authors.

**Figure 2 pharmacy-09-00039-f002:**
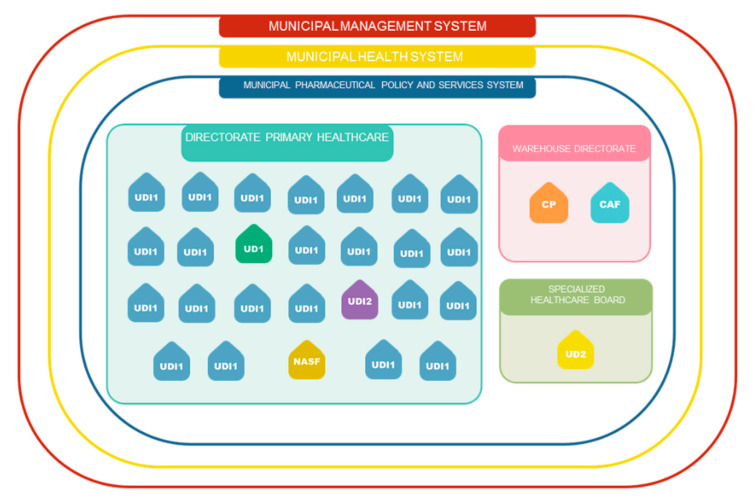
PPS organizational structure. Legend: UDI1: Dispensing unit in PHC unit; UDI2: Dispensing unit in specialized care service (medications for STI / AIDS, leprosy, tuberculosis, and other strategic control diseases for Ministry of Health); UD1: Central Pharmacy—PHC medications dispensing center; UD2: Specialized and Judicial Dispensing Unit (mainly high-priced drugs, for rare or chronic diseases or for judicial access); CP: Purchasing department; CAF: Pharmaceutical Distribution and Supply Center; NASF: Family Health Support Center (Ministry of Health program to support PHC). Source: The authors.

**Table 2 pharmacy-09-00039-t002:** Environmental (Scenario) and the internal systems components of the Sociotechnical System of PPS in primary healthcare (PHC).

Components	Literature Description	Adaptation to the Study
Components		Components of External System: Municipal Management System; Municipal Health System
Environment	Context where the study’s focus system is inserted.	Characterization of the environment where PPS is inserted according to: Objective, guidelines, and society’s goals for PPS.
Interested parts	Objective of the system for patients, management, suppliers, and representatives of civil society.	PPS expected outcomes from the perspective of municipal management, the health department, and patients.
Financing	Economic situation under which the system is developed and what financial resource the system should operate.	Annual financing of PPS. Budgeted amount and amount paid.
Regulations	Rules and laws that regulate the organization.	Rules and municipal laws that regulate the organization and activities of PPS.
Components		Components of the Internal System
Goals	Targets of the system, its operation, or that generated the demand for its construction.	Objective of municipal PPS for the different healthcare actors.
Management	Organizational structure and its technical operations.	Organizational structure of municipal PPS and its management capacity in PHC. Dimensions: Organizational, operational, and sustainability.
Workforce	Number of people able to participate in the social division of labor process.	Group of people with the capacity and ability to carry out PPS activities. Work groups where tasks are performed.
Structure	Equipment or the physical structure required for the performance of system activities.	Structures and infrastructure by workgroup of the PPS system.
Processes	Main activities that are part of the system, including the main and routine activities.	Main activities by PPS workgroup in PHC. Why does the unit of work exist? How does it communicate with the system and the environment in its technical and relational aspects?
Technology	Equipment and methods used to produce products or services. A health technology can be defined as a way, knowledge, and the instruments used to produce health actions.	Description of drug treatments and instruments used for their access, use, and monitoring. Drug treatment: Criteria and places for selection, access, and monitoring of outcomes. Instruments: Municipal List of Medicines, Electronic Health Record, protocol, computerized system.
Culture	Organizational culture of the system, the way it thinks and acts, the beliefs and values held in organization and society.	Organizational culture of PPS on three aspects. Symbols: Names, logos, and physical characteristics used to convey the organization’s image. Rituals: Usual and repeated actions within an organization. Ideology: Beliefs, moral principles, and values provide the basis for organizational decision making.

**Table 5 pharmacy-09-00039-t005:** Description of the purpose and goal for PPS in the municipal health plans (MHP) 2010–2013 and 2014–2017.

Period	Purpose	Goal
2010 to 2013	“The municipal PPS’s main purpose is to provide regular and free supply of selected drugs, in order to contemplate the outpatient drug treatment of the different pathologies that affect the population”.	Ensure the supply of medicines to all SUS patients in accordance with the list of selected medicines. Constantly review the list of selected medicines. Deploy Phytotherapic Handling Pharmacy.
2014 to 2017	“Guarantee the dispensation medicines of municipal list to the population”. Implement the herbal medicine production laboratory.	Consolidate drug purchase, dispensing and inventory control routines, and the use of the G-MUS management system to improve purchase and dispensing monitoring. Forward a fundraising project at the Ministry of Health to implement the herbal medicine laboratory.

Source: MHP 2010–2013 and 2014–2017.

**Table 6 pharmacy-09-00039-t006:** Summary: Assessment matrix of the management capacity of municipal PPS—diagnostic stage in 2017.

Description	MS	SO
**Management component**		
Organizational	96	16
Operational	100	49
Sustainability	96	60
**Total**		
**Workforce component**		
Condition of existence of the PPS Coordination in the Municipal Health Department (MHD).	5	0
Degree of decision-making autonomy of the PPS Coordination.	10	0
Profession of the PPS coordinator.	7	0
Participation of pharmacists in the preparation of the Municipal Health Plan.	10	0
Participation of PPS coordination in health programs or activities in the municipality (in other MHD sectors).	10	0
Responsibility for defining the programming parameters for distributing medicines to health units.	7	0
Health units with pharmacists working in the team.	7	1
Pharmacists trained in PA, management, public health, or related fields in the past 2 years.	7	2
Type of employment contract of the PPS coordinator.	7	0
Pharmacist position in the list of municipal public service positions.	7	7
**Total**	77	3
**Infrastructure component**		
Integration of the PA information system with that used in the health care network.	6	6
Instruments to assess physical and environmental conditions to store medicines (external and internal conditions, lighting, refrigeration, security).	6	3
Investments in infrastructure in the last 4 years in PPS services.	7	7
**Total**	19	16
**Processes component**		
Pharmacists and health unit coordinators recognize the existence of PPS coordination.	8	0
Regular functioning of the Pharmacy and Therapeutics Committee in the last year.	6	0
Pharmacists know the Municipal Health Plan.	8	8
PHC drugs purchased based on the schedule.	6	0
Prescribed medicines based on lists of medicines adopted by the municipality.	6	6
Medicines out of date available for dispensing.	6	6
Health services have a waste management plan.	5	0
The pharmacist’s productivity record has a defined procedure.	7	0
Spending on medicines to meet legal demands in relation to the budget to buy medicines, in the last year.	6	6
Schedule for regular distribution of medicines to health units: Monthly, biweekly, or weekly.	6	6
Prescribers receive information about the availability of medicines in pharmacies at health facilities.	7	7
Prescribers know how to find updates to the municipal list of medicines.	8	8
Amount of medications available in health facilities suitable to meet patients’ demands (perceptions of health secretary, PA coordinator, pharmacists, and prescribers).	4	1
Diversity of medications available in health facilities suitable to meet patients’ demands (perceptions of health secretary, PA coordinator, pharmacists, and prescribers).	4	2
Procedures for monitoring the PPS and using data to plan actions.	6	0
Resources diversity introduced in the municipal PPS in partnership with the State PPS.	7	0
Shared definitions about goals, guidelines, and targets PPS in the Municipal Health Plan.	10	0
Official means to receive criticisms and suggestions about medicines from patients (referred by the health unit coordinator).	7	7
Official means to receive criticisms and suggestions about medicines from patients (referred by patients).	8	3
**Total**	125	70
**Technology component**		
Pharmacy and Therapeutics Committee has formal institution at the MHD.	5	0
Technical criteria used to prepare the medication schedule.	6	0
Municipal Medicines List Availability.	6	5
Municipal List of Medicines includes drugs are outside National List of Essential Medicines or lists agreed in SUS.	5	5
Instruments to standardize medication dispensing (SOP, manuals).	6	0
**Total**	28	10
**Organizational culture component**		
Communication strategies between health units and PPS coordination to resolve medications issues (referred by health unit coordinators and pharmacists).	7	7
Articulation between PPS coordination, the Family Health Strategy coordination, and Community Agents Team.	7	0
Partnership between the municipality’s PPS Coordination and the State PPS Coordination.	7	0
PPS agenda at the meetings of the Municipal Health Council in the last 4 years.	10	3
**Total**	31	10

Source: The authors. Legend: MS: Maximum score; SO: Score obtained.

**Table 7 pharmacy-09-00039-t007:** Number of attendance in dispensing services, items dispensed, and its price in 2017.

Description	Patients	Number of Visits	Number of Items	Quantity of Dispensed Items	Values of Total Dispensed Items (R$)
Daily average ^1^	928	971	145	71,043	8375.00
Monthly Average ^2^	20,406	21,355	3193	1,562,936	184,254.00

Source: The authors. ^1^—daily average calculated based on the data of dispensation of July 1st days to December 31st of 2017; ^2^—monthly average calculated based on six month dispensation data.

## Data Availability

All data are available in the links and references indicated in the article.

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
