# Peer review of "A Sociotechnical Approach to Analyze Pharmaceutical Policy and Services Management in Primary Health Care in a Brazilian Municipality"

_pharmacy, 2021, doi:10.3390/pharmacy9010039_

Round 1

Reviewer 1 Report

Thank you for the opportunity to review your manuscript.  I find your work, attempting to integrate the concerns and frustrations of a variety of entities working with the Pharmaceutical Policy and Services System fascinating.  

I have a general and overriding comment for you as you work to revise your submission - - have someone who does NOT understand the process (methods) read your work and then tell you what you did.  I believe you will find that there are gaps in the logic, that are evident to you that do not translate into the manuscript.  I personally struggled to find some of the linkages.

I also have some specific comments:

Line 3:  The spelling of management in the title is non-standard

Line 10:  I do not know what "stabilishing" means

Line 16:  The sentence starting with "The lack..." is incomplete or confusing.  perhaps the word "and" in place of the comma would make it complete, but I do not know that it would have your intended meaning.

Line 25:  While I appreciate that the sociotechnical analysis is the key to your work, I don't know that it's a new level of "development".  It may be the development of a new level of analysis, or something like that.

Line 36: I'm confused as to what "it" is.

Line 67:  This sentence doesn't make any sense.  Your premise is that because the PPS system is so highly complex that it should be evaluated using your sociotechnical system, but the "phenomenon" of the "operation" confuses me.

Line 87: interprofessional collaboration is 2 word and most often written inter-professional collaboration.

Line 88:  it is not standard, and in fact disallowed in pharmacy to have a leading zero in a whole number.  This happens many times in this manuscript. This should read "in PHC for four municipalities..."

Line 95:  135,000  not 135 thousand

Line 122:  list the YouTube video as a reference at the end of the manuscript, do not embed it in your manuscript.

Line 134:  Pharmacies located AT or IN municipal health departments

Line 138:  more leading zeros that must be removed

Line 150:  This figure is nearly impossible to read.  Consider printing the titles in BLACK rather than white for Municipal Management System and Absolutely for the 2nd item with the yellow background - I was never able to find a way to read that.

For the slices of the pie chart, also consider making the words readable by using BLACK.

I believe the word structure in the green pie slice is spelled wrong.

Line 187: the table, 1st article narrative, there needs to be a space between missing and from in line 4

Line 204:  This table has awkward spacing, look at the budget figure 2,034,534.40. that needs to be all on 1 line

Line 232:  Again, a figure that is barely readable. The white lettering makes it impossible to determine what you are trying to say.  Further the coloring of the unit boxes isn't clear.  Why, for instance, is there a green UD1 box?  no description if the colors is given in the legend.

Line 249:  more leading zeros

Line 256: Service Center is 2 words

Line 260:  should be plural pharmacists

Line 328:  another leading zero

Line 353:  four indicators, no leading zeros

Line 355:  again

Line 379:  The results of your work revealed something, but your work is NOT an article, it is so much more than that.

Line 399:  of the PPS architecture

Author Response

Thank you very much for your careful review, we greatly appreciate your collaboration.

I have a general and overriding comment for you as you work to revise your submission - - have someone who does NOT understand the process (methods) read your work and then tell you what you did.  I believe you will find that there are gaps in the logic, that are evident to you that do not translate into the manuscript.  I personally struggled to find some of the linkages.

Thank you for the great suggestion. We asked someone else and we then tried to clarify the Methods section.

 I also have some specific comments:

Line 3:  The spelling of management in the title is non-standard – Sorry, we could not find the error.

Line 10:  I do not know what "stabilishing" means. The sentence has been rewritten.

Line 16:  The sentence starting with "The lack..." is incomplete or confusing.  perhaps the word "and" in place of the comma would make it complete, but I do not know that it would have your intended meaning. Yes, the “and” was necessary.

Line 25:  While I appreciate that the sociotechnical analysis is the key to your work, I don't know that it's a new level of "development".  It may be the development of a new level of analysis, or something like that. The phrase was rewritten, accepting the suggestion.

Line 36: I'm confused as to what "it" is. It was rewritten “Pharmaceutical policies cannot be based…”

Line 67:  This sentence doesn't make any sense.  Your premise is that because the PPS system is so highly complex that it should be evaluated using your sociotechnical system, but the "phenomenon" of the "operation" confuses me. Thanks for the suggestion. The sentence has been rewritten.

Line 87: interprofessional collaboration is 2 word and most often written inter-professional collaboration. Changes made.

Line 88:  it is not standard, and in fact disallowed in pharmacy to have a leading zero in a whole number.  This happens many times in this manuscript. This should read "in PHC for four municipalities..." Changes made.

Line 95:  135,000  not 135 thousand Change made.

Line 122:  list the YouTube video as a reference at the end of the manuscript, do not embed it in your manuscript. Transferred to References

Line 134:  Pharmacies located AT or IN municipal health departments. Change made.

Line 138:  more leading zeros that must be removed. Changes made.

Line 150:  This figure is nearly impossible to read.  Consider printing the titles in BLACK rather than white for Municipal Management System and Absolutely for the 2nd item with the yellow background - I was never able to find a way to read that. Change made.

For the slices of the pie chart, also consider making the words readable by using BLACK.

I believe the word structure in the green pie slice is spelled wrong.

Line 187: the table, 1st article narrative, there needs to be a space between missing and from in line 4. Change made.

Line 204:  This table has awkward spacing, look at the budget figure 2,034,534.40. that needs to be all on 1 line. Spacing changed.

Line 232:  Again, a figure that is barely readable. The white lettering makes it impossible to determine what you are trying to say.  Further the coloring of the unit boxes isn't clear.  Why, for instance, is there a green UD1 box?  no description if the colors is given in the legend. Changes made. Change made.

Line 249:  more leading zeros. Removed.

Line 256: Service Center is 2 words. Change made.

Line 260:  should be plural pharmacists. Change made.

Line 328:  another leading zero. Removed.

Line 353:  four indicators, no leading zeros. Removed.

Line 355:  again. Removed

Line 379:  The results of your work revealed something, but your work is NOT an article, it is so much more than that. You are right. The project revealed something.

Line 399:  of the PPS architecture. Change made.

Reviewer 2 Report

It is well known that preventive measures have the greatest impact on health of the population. However, it should be remembered that medical interventions, including pharmacotherapy are also important. Each citizen has been, is or will be a patient. Role policy makers are to ensure that health systems are operational and that helps prevent disease and, in the event of an illness, provide the best possible care and pharmacological treatment. Drug policy is an integral part of health policy. The desire is to the implemented state drug policy was consistent with international recommendations for ensuring the widest possible human rights for treatment. Drug policy holds also a much wider, economic dimension, due to the important role of industry as a partner in the quest to improve life expectancy and conditio citizens' health. Therefore, the subject of presented manuscript is important and current. However before publications, the Authors should consider and improve the following points:

  • The title should contain information that the research carried out relates to the situation in one of the municipalities in Brazil.
  • There is no introduction in the abstract. The first sentence is incomplete, it should be corrected. The abstract should be improved, it should contain the same elements as the manuscript in the shortened form.
  • In the introduction, it is worthwhile to describe in more detail the issues raised in lines 60 to 66.
  • Why was only 1 pharmacist involved in the data collection described in section 2.2.2. In my opinion, this is a completely unjustified failure.
  • The discussion should also include a paragraph in which the Authors will suggest ways to improve the current situation.
  • In my opinion, a large amount of non-English literature makes difficult to understand the problem.

Author Response

Thank you so much for your attention. Please see below our responses.

  • The title should contain information that the research carried out relates to the situation in one of the municipalities in Brazil.  - We included “a Brazilian municipality”.
  • There is no introduction in the abstract. The first sentence is incomplete, it should be corrected. The abstract should be improved, it should contain the same elements as the manuscript in the shortened form. - The introduction was included. We have attempted a way to describe all the elements included in the results, however, it resulted impossible. With the limited number of words in the abstract, we decided to highlight the most important elements of them system, that characterizes it. We have rewritten some parts only.
  • In the introduction, it is worthwhile to describe in more detail the issues raised in lines 60 to 66. - We attempted to express the counterpart of the problems, highlighting the social part of the system is usually missing in the studies in this field. The technical problems are already listed in the paragraph.
  • Why was only 1 pharmacist involved in the data collection described in section 2.2.2. In my opinion, this is a completely unjustified failure.   - The Assessment Matrix of the Management Capacity of municipal PPS is composed of a set of indicators of the governance/management, operability and sustainability of the PPS. In its data collection methods it is required that different sectors’ representatives answer different questions. The questions that must be answered by pharmacists are related to infrastructure, official rules, usual procedures and documents related to the pharmaceutical system in that municipality. It does not change between the pharmacists enrolled because it not about their view, it is about the PPS it self. In this municipality size, the method does not require more than one pharmacist to answer these questions. So, the pharmacist enrolled in the pharmacy of the Health Department was recruited to answer about general data. In the other hand, the questions answered by nurses, doctors, assistants and patients are related to their perceptions of the pharmaceutical system, so a bigger sample is need. All the 6 pharmacists employed in this municipality health system were engaged in the other data collection phases (through the testimonies in the documentary adopted as one data source). Quotes of these testimonies can be find in the results section.

  • The discussion should also include a paragraph in which the Authors will suggest ways to improve the current situation. -  Thanks for the suggestion. We made some changes in the text to comprehend the suggestions.
  • In my opinion, a large amount of non-English literature makes difficult to understand the problem. - We agree. In fact, the scientific literature about the pharmaceutical policy and practices in Brazil is mainly published in Portuguese, in national journals. The data sources for this study are all in Portuguese only. We believe it is important to increase the publication of the Brazilian experience of pharmaceutical policy and practices in public health because of its relevance in the global scenario. Doing so, we believe the dialogue between the Brazilian and foreign researchers will be easier and fruitful.